# Mechanism of DNA origami folding elucidated by mesoscopic simulations

Marcello DeLuca[1], Daniel Duke [1], Tao Ye [2,3], Michael Poirier[4], Yonggang Ke [5], Carlos Castro [6] & Gaurav Arya [1] ✉

Many experimental and computational efforts have sought to understand DNA origami folding, but the time and length scales of this process pose significant challenges. Here, we present a mesoscopic model that uses a switchable force field to capture the behavior of single- and double-stranded DNA motifs and transitions between them, allowing us to simulate the folding of DNA origami up to several kilobases in size. Brownian dynamics simulations of small structures reveal a hierarchical folding process involving zipping into a partially folded precursor followed by crystallization into the final structure. We elucidate the effects of various design choices on folding order and kinetics. Larger structures are found to exhibit heterogeneous staple incorporation kinetics and frequent trapping in metastable states, as opposed to more accessible structures which exhibit first-order kinetics and virtually defect-free folding. This model opens an avenue to better understand and design DNA nanostructures for improved yield and folding performance.

DNA nanotechnology forgoes DNA's conventional use as an information storage medium and instead uses it as a structural material, manipulating the canonical base-pairing rules to fold DNA into functional structures[1–7] and dynamic devices[8–11]. A widely used design paradigm for creating such structures is DNA origami[3]. The self-assembly process of DNA origami involves the hybridization of hundreds of individual oligonucleotide "staples" to one or multiple[12,13] much longer "scaffold" DNA strands; the staples constrain the scaffold by hybridizing to it at two or more separate regions, effectively "folding" the scaffold into a quasi-2D or 3D shape. The intended outcome of this folding process is a shaped structure with a single minimum-energy conformation in the case of static structures, or two or more minimum-energy conformations in the case of dynamic structures[8]. However, the yield of this folding process can vary widely, and the process itself can sometimes completely fail, especially for larger structures, presumably due to the presence of kinetically trapped states. A better understanding of the self-assembly process would enable scientists to create structural designs that fold faster and with

better yield and could also enable new dynamic function in actuation or in competitive assembly. This process is also fundamentally compelling as the folding mechanisms of DNA nanostructures are very interesting on their own and may be relevant to various other self-assembled systems such as proteins[14–16], polymers[17], and colloids[18].

Experimental efforts to study origami folding can be broadly categorized into two groups: direct imaging and fluorescence approaches. Atomic force microscopy (AFM) is typically used for direct imaging and is able to resolve individual origami species, providing insight into the folded character of structures[19–23]. Using this approach, researchers have shown that the folded product of DNA origami can be significantly influenced by design and fabrication conditions such as staple sequences[24], ionic conditions[25], and staple-scaffold stoichiometry[26]. While useful, AFM can provide only limited information because it has relatively low temporal resolution. It is also challenging to acquire high-resolution AFM images of partially formed origami, although this technique is beginning to see use in probing folding pathways[23]. Fluorescence measurements are capable of

[1]Thomas Lord Department of Mechanical Engineering and Materials Science, Duke University, Durham, NC 27705, USA. [2]Department of Chemistry & Biochemistry, University of California, Merced, CA 95343, USA. [3]Department of Materials and Biomaterials Science & Engineering, University of California, Merced, CA 95343, USA. [4]Department of Physics, The Ohio State University, Columbus, OH 43210, USA. [5]Department of Biomedical Engineering, Georgia Institute of Technology and Emory University, Atlanta, GA 30322, USA. [6]Department of Mechanical and Aerospace Engineering, The Ohio State University, Columbus, OH 43210, USA. ✉e-mail: gaurav.arya@duke.edu

tracking individual staple incorporation in DNA origami and have been used to study folding thermodynamics and kinetics[19,26,27]. A recent study expanded this approach to many staples, generating fluorescence curves from each individual staple to gain a deeper understanding of the order of staple incorporation[28]. This study revealed that structures can fold with multiphase kinetics and that DNA origami may follow highly heterogeneous folding pathways. However, because fluorescence measurements report bulk signals from all possible pathways, individual folding pathways remain concealed. Overall, efforts for investigating DNA origami folding are hindered by a lack of experimental techniques that can resolve individual events or provide adequate spatial resolution for folding structures.

Computational modeling offers a viable microscopic route to study DNA origami folding and its dependence on design parameters. One intuitive approach is to represent DNA at the binding domain level and treat hybridization using chemical kinetics, where sequence-based free energy differences are used to determine transition rates between intermediate states. Kinetic simulations of a network of such intermediate states then allows the most likely folding order of structures to be predicted[29]. However, each hybridization event in the folding process is not only coupled to other hybridization events, but there is also a strong stochastic component to the binding order, and kinetic rates from one hybridization state to another likely depend on the order of hybridization. Thus, proper treatment may require accounting for not only the Markovian transition rate from one state to another (i.e., assuming that the transition rate to the next state is dependent on just the current state) but potentially every transition rate from every state to every other state including the binding order to accurately capture overall folding rate and yield, which would be intractable. Lattice-based Monte Carlo simulations can partly alleviate this problem because they retain configurational information, and this approach has indeed been used to uncover details of origami folding, specifically nucleation processes[30]. However, these simulations are still quite coarse and cannot capture the dynamic behaviors of the staples and scaffold, namely their diffusion, conformational flexibility and transitions, and relative timescales of these processes, all of which are critical to origami folding.

Dynamic modeling of DNA origami folding[31] suffers from the same scale problem as the more ubiquitous problem of modeling protein folding dynamics[16]. These are both complex multi-step processes with potentially long transition times between states. All-atom molecular dynamics simulation is a common technique for direct simulation of DNA and has been used to accurately capture the behavior of small DNA motifs[32], local dynamics of DNA nanostructures[33,34], and effects arising from specific ions[35,36]. However, the timescale access of these simulations is too limited to capture the self-assembly of even the smallest DNA nanostructures. Coarse-grained (CG) models such as oxDNA[37–39] can access much longer timescales at the expense of reduced detail, and these models are routinely used to study dynamics of large pre-folded DNA nanostructures[40] as well as hybridization-related processes[41]; the oxDNA model has even been used to probe the self-assembly of a small, highly-accessible DNA origami structure[42]. However, existing CG models are unable to capture the entire process of DNA origami assembly for structures larger than a few hundred nucleotides, and the computational cost of folding even small DNA nanostructures is essentially prohibitive for investigating bulk quantities like kinetic rate constants, which would require many repeats of these simulations to be performed.

Overall, the prohibitive timescale of DNA origami folding has made its direct simulation elusive for structures of practical sizes. Further coarse-graining appears to be an obvious solution, but a representative particle the same size as a single nucleotide or smaller is needed to capture the behavior of both single-stranded DNA (ssDNA) and double-stranded DNA (dsDNA). This is because the hybridization of two strands into a duplex must also implicitly capture the change in

bending properties of the newly formed double-helix through base-pairing and base-stacking interactions across the helices. To coarsen a model such as this further would sacrifice the geometric interactions that allow the properties of dsDNA to naturally and correctly emerge upon hybridization without actively modifying the underlying DNA force field. Further coarsening would thus severely compromise the accuracy of the model.

Here, we present a mesoscopic model that addresses this issue of correctly capturing the mechanical properties of DNA in different hybridization states with a viable CG unit that allows for long-timescale simulations (Fig. 1). Our model represents DNA at a resolution of 8 nucleotides per bead. It uses common interaction potentials for backbone connectivity, bending, and excluded volume, and employs a parameterized hybridization potential. A key feature of this force field is that it automatically switches between the properties of ssDNA, dsDNA, and other motifs like Holliday junctions depending on whether or not the DNA is identified as hybridized. Brownian dynamics simulations conducted with this switchable force field model allows us to capture the entire folding process, from dissociated free-floating species to fully assembled origami structures. By studying several DNA origami designs with these simulations, we were able to uncover the dynamic mechanism of DNA origami folding and to reveal differences in the folding behavior of different designs and their permutations, including variations in scaffold routing and staple design.

## Results
### Model development
Our first task was to choose a target representation for DNA, which is treated using a mesoscopic bead-chain model (Fig. 1a). A convenient domain size in DNA origami is seven nucleotides for designs using a honeycomb lattice or eight nucleotides for designs on a square lattice. These domain sizes allow for DNA crossovers to be located at even multiples of seven or eight nucleotides from each other. For simplicity, we only considered square lattice structures in this study, and thus, the representation was selected as 8 nucleotides per bead. However, the model can be readily translated to simulate designs on a honeycomb lattice. To determine whether this coarsening level could be sufficient to capture the conformational dynamics of typical DNA nanostructures, we compared principal component analysis (PCA) of oxDNA[38] simulations of a pre-assembled DNA origami structure using individual nucleotide coordinates to PCA of the same simulations using coordinates which were coarsened to match our target representation (centroids of every eight contiguous nucleotides). We chose a test sheet-like structure inspired by a structure previously studied by ref. 42 for this analysis because it is known to be highly dynamic and exhibits several distinct dynamic modes. We found that all 21 relevant principal components accounting for >99% of total structural fluctuations were reproduced (see Methods and Supplementary Figs. 1, 2), indicating that our eight-nucleotide coarse-grained representation may have the capacity to fully capture the structural dynamics of DNA origami.

Our model uses standard potentials for treating excluded volume interactions between beads (short-range repulsive potentials[43]), inter-bead spacing along the DNA backbone (harmonic potentials), and backbone bending (harmonic potentials) based on experimentally measured properties of DNA (see Methods and Supplementary Fig. 3). The hybridization potential was derived from the oxDNA simulations[38]: the potential of mean force (PMF) of a pair of eight-nucleotide fragments was computed as a function of separation distance between their centroids using umbrella sampling at 300 K and 0.5 M effective monovalent salt concentration, consistent with typical origami folding conditions. Removing the translational entropy contribution from the PMF and linearizing the resulting profile yielded a tangible approximation of the inter-bead hybridization potential that can be included in our model to reasonably reproduce the hybridization

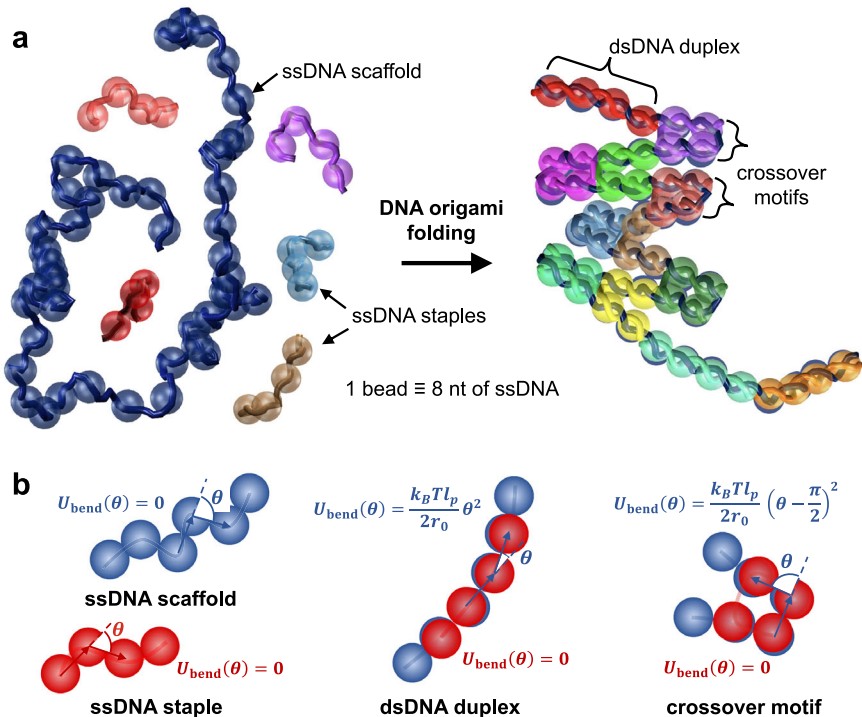

**Fig. 1 | Mesoscopic representation of DNA origami in our model. a** Fully disassembled and assembled structures before and after folding. Not all staples are shown for clarity. The underlying DNA strands are shown as helical tubes for reference. The beads used to represent DNA are shown in dark blue for the scaffold and in different colors for each staple oligonucleotide. **b** Switchable force field used for capturing transitions between single-stranded DNA, double-stranded DNA, and crossovers.

behavior of DNA at this coarseness level (see Methods and Supplementary Fig. 4). We employ pairing matrices to enable hybridization between specific beads; for simplicity, we do not allow misbinding to occur, and only perfectly complementary scaffold-staple bead pairs (i.e., complementary scaffold-staple segments that should be paired in the target structure) are allowed to bind. All eight-nucleotide domains bind with the same binding enthalpy (depth of hybridization potential) regardless of hypothetical sequence. Differences in binding strength due to domain lengths differing by at least eight nucleotides (e.g., 8 vs. 16 nucleotide domains) are captured implicitly. This binding is reversible, so pathways requiring misbound species to dissociate for folding to complete can be properly represented. The reversal of this binding occurs at a rate dictated by the hybridization strength (melting barrier) of 10 kcal/mol per bead (see Methods).

A key feature of this model is that the force field can induce a modification of the bending and backbone separation potentials of the scaffold and staples following hybridization (Fig. 1b). For example, if three or more consecutive beads of DNA scaffold become bound (based on distance criteria where complementary species closer than 2 nm are considered bound, see Methods) to three or more staple beads, a bending potential is activated to enforce the persistence length of dsDNA in only this region. Unhybridized portions of the scaffold elsewhere will continue to conform based on the behavior of ssDNA, where no bending potential is enforced since the bead size is on the order of the Kuhn length of ssDNA (~3 nm at the salt concentrations typical of origami assembly conditions[44]). Crossovers in DNA origami design also exhibit switchable behavior, with their bending potentials being activated when enough portions of the crossover (three consecutive connected beads, of which at least two constitute the crossover) become bound; beads identified as part of a crossover assume a 90-degree equilibrium bending angle when bound instead of the zero-degree bending angle that would apply to dsDNA, thereby enforcing the correct conformations of this motif.

Because the beads in this system are sufficiently large (and thus have a sufficiently large relaxation time), we can use overdamped Langevin (Brownian) dynamics[45,46] to efficiently integrate their equations of motion (see Supplementary Discussion 1 and Methods). To this end, we developed a Brownian dynamics simulation software package that implements the model introduced above and can be used to conduct mesoscopic simulations of DNA origami folding.

## Model validation

To test whether this model can capture the dynamics of DNA nanostructures correctly, we simulated the pre-assembled representative sheet structure using oxDNA[38], coarsened the trajectory to match our model's representation, and conducted PCA on the coarsened trajectory. With our coarsened oxDNA trajectory as a reference, we then simulated the same structure using our mesoscopic model. To compare the mean structures captured by the two models, we used a proximity mapping approach whereby a distogram was constructed describing the average separation distance between each pair of eight-nucleotide regions in the structure for both our model and for oxDNA. As is evident from Fig. 2a, the maps corresponding to the two models are very similar, with our model slightly underestimating the end-to-end distance of the structure owing to a small amount of additional rotational flexibility in crossovers resulting from the lack of torsional potentials in our model, a compromise that we made to improve model simplicity. Nevertheless, our model still captures the mean structure quite well.

To investigate if the model can also capture the overall conformational dynamics (fluctuations) of the structure, we then carried out PCA of the trajectory generated using our model and compared the results to PCA of the coarsened oxDNA trajectory (Fig. 2b). We found that the first three PCs are reproduced, though the first two principal components of the trajectory from our mesoscopic model are reversed in order of importance compared to the coarsened

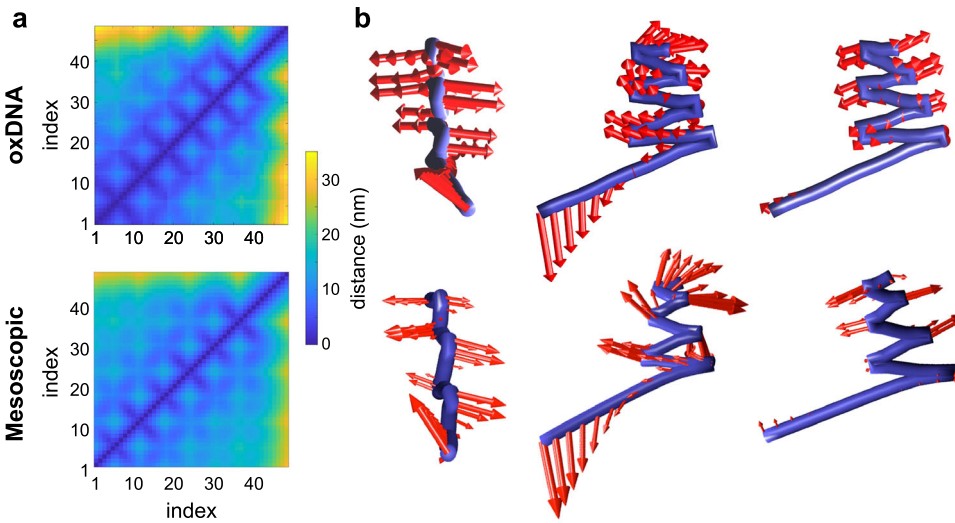

**Fig. 2 | Model validation against oxDNA. a** Distograms from oxDNA simulations of a sheet-like structure (768 nt/500 kDa) and from simulations of the same sheet-like structure with our mesoscopic model. The oxDNA simulations were coarsened to the same representation as our model. "index" refers to individual eight- nucleotide sections of the scaffold. **b** First three principal components of motion from oxDNA and from our model. Note that the first two principal components from our model are reversed in dynamic importance (Supplementary Fig. 9).

oxDNA trajectory. Overall, both simulations have 21 similar relevant PCs accounting for >97% of structural fluctuations, and both structures exhibit similar root mean structure fluctuations (Supplementary Fig. 10). Other basic features like end-to-end distance of ssDNA and dsDNA pieces and the distribution of spacing between continuous duplex sections and crossover sections of the scaffold were also validated against oxDNA simulations (Supplementary Figs. 6–8). Thus, our model captures the conformational ensemble of dsDNA, ssDNA, crossovers, and entire DNA origami structures quite well.

### Dynamics of DNA origami folding

Having validated our model, we next used it to simulate DNA origami self-assembly. We began by studying the folding of a simple four-helix bundle (4HB) (designs can be found in Supplementary Figs. 11–15). The simulations were carried out at a constant temperature of 300 K to mimic isothermal folding conditions. To quantify the kinetics of staple binding, we examined the concentration $C$ of free (unhybridized) staple domain*s* over the course of the simulation. We used identically sized staple species throughout the structure so that any observations about the folding pathway are purely dependent on the scaffold routing and geometry of the final structure. The slope of the natural log of the ratio of the current free staple domain concentration to the initial free staple domain concentration, $\ln(C/C_0)$, plotted against time describes the overall rate constant of staple binding to the scaffold. A linear kinetic curve would indicate that the reacting species are exhibiting first-order kinetics and nonlinear curves would either indicate that the reaction is first-order but contains a changing rate constant in time or that higher-order kinetics are at play.

We began by studying the folding of a 4HB structure with a circular scaffold routing that runs straight across the structure with only four total scaffold crossovers and a repeating modular staple pattern that is uniform across the entire structure to remove staple variability as a consideration in the assembly process (Fig. 3a and Supplementary Fig. 11). We first simulated the folding process with our derived model parameters, using a hybridization enthalpy of −10 kcal/mol as obtained from our parameterization. Upon simulating this structure's self-assembly ten times and computing the average kinetic behavior, we found that it exhibits two distinct first-order regimes of folding (blue curve, Fig. 3b and Supplementary Fig. 17). The first-order kinetics indicate that this design follows an indeterminate folding path, where

species bind to the structure with little concern for order. This makes intuitive sense, as there is nothing preventing these very similar staple domains from diffusing to one binding site over another, and they all exhibit identical hybridization enthalpies with their complements.

To understand the observed change in kinetic rate during assembly, and, more generally, the folding process, we produced visualizations of the structure as it was folding (Fig. 3e; also see Fig. 3f for a schematic depiction of the key events involved in this folding process and Supplementary Movies 1–4 for animations of the simulated trajectories). The scaffold begins in an open, disordered configuration reminiscent of a swollen circular polymer in good solvent[47]. The first portion of the folding process involves staples diffusing to the scaffold and binding with one of their domains. Upon binding, each staple enters a hyperlocal search along the scaffold for additional complementary domains and typically finds one additional domain to bind to, thereby bridging the scaffold at those two points. This proceeds for some time, with many staples binding at two locations and the scaffold establishing a long, two-stranded character, but these staples typically have an additional domain which is unable to find its complement, likely due to the large energetic penalty of bending the scaffold to establish an additional contact. This is evidenced by the fraction of each incorporated staple which is bound to the scaffold, $f_b$, stalling around a value of 0.5, which indicates that, on average, two domains in each incorporated staple are bound, and two remain unbound (Fig. 3d inset). Because the scaffold is highly accessible, all staples bind with the same rate constant and so we observe this as the initial faster first-order process.

At some point, with remarkable consistency, this 4HB structure experiences a collapse event whereby the scaffold transitions from the two-stranded structure established in the first phase of folding into a "proto rod" structure, possessing roughly the shape of the final structure but significantly swollen and shorter, and lacking global order. This transition appears to occur via a cooperative zipping process mediated by partially bound staples resolving their third connection to the scaffold in a successive manner along the length of the scaffold, starting where the scaffold bends to accommodate this change (Fig. 3d inset and 3e, f). It is possible that this zipping motion is a critical event that is required to overcome the penalty of constraining the partially assembled scaffold to itself. Furthermore, it is noted that the loops formed opposite the zipping boundary are sterically repelled

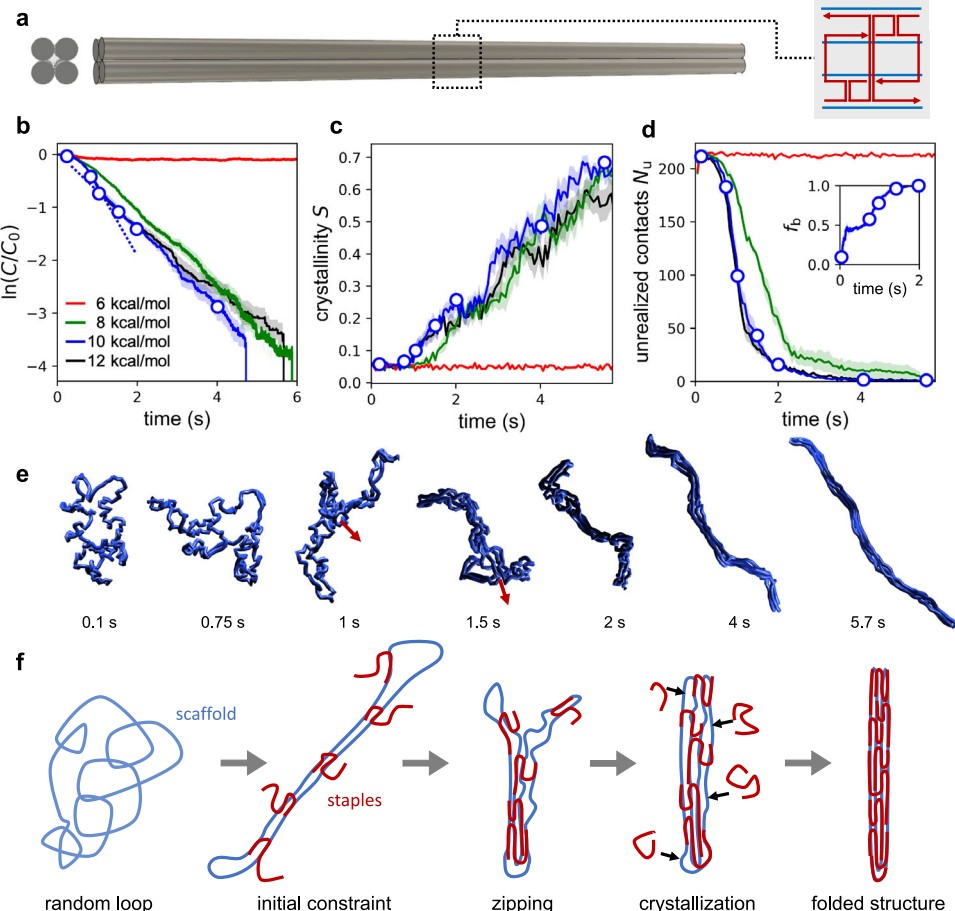

**Fig. 3 | Folding behavior of a 4HB structure and its dependence on staple binding strength. a** Design of the 4HB structure (3584 nt / 2.33 MDa). Staples follow a repeating pattern, as shown on the right. **b–d** Dependence of the folding mechanism on the strength of staple binding 6, 8, 10, and 12 kcal/mol), as characterized by the variation of three quantities as a function of time over the course of the folding simulation: staple concentration normalized by their initial concentration (**b**) which is observed to experience a change in slope between 0.7 and 1 s, Landau-De Gennes crystallinity parameter describing global order (**c**), number of unrealized contacts during folding describing local order (**d**), and the mean fraction of incorporated staple strands which are bound to the scaffold (**d,** inset). The plotted lines represent averages of data collected from ten independent simulation runs and the shaded envelopes surrounding those lines represent SEMs. **e** Representative images of scaffold conformations during assembly. These correspond to the open circles shown in panels **b–d**. Arrows represent zipping direction. **f** Schematics illustrating the hierarchical origami folding process.

and thus splay, hence the tendency for only a single side to begin zipping (Fig. 3e, red arrows). It is our interpretation that the increased excluded volume surrounding the binding sites following collapse into the proto rod (thereby reducing the accessibility of binding sites) is responsible for the subsequent reduction in kinetic rate, whereby the folding proceeds as a slower first-order process until the final staple binds and forms the completed structure.

To better characterize this collapse mechanism, we computed the time-resolved Landau-de Gennes parameter $S$, which describes the crystallinity of structures, that is, the global alignment of backbone tangent vectors, where $S = 1$ corresponds to a completely straight backbone and $S = 0$ a completely disordered chain (see Methods). Although the structure crystallizes significantly by the end of the simulation, the observed collapse into a proto rod does not correspond to a significant increase in crystalline order (Fig. 3c); the structure instead collapses into a locally ordered but globally disordered structure, after which crystallinity is established gradually as additional staples bind. This is corroborated by calculation of a parameter $N_u$ which describes the number of nonbonded scaffold neighbor contacts that have not yet been realized in the structure (see Methods). Our results show that almost all nonbonded neighbors make contact very early during folding, with little improvement afterward (Fig. 3d), while the crystallinity parameter increases gradually but significantly

(Fig. 3c). This indicates that this structure establishes local order before it establishes global order; the severity with which this occurs should depend on the staple to scaffold ratio.

## Dependence on hybridization enthalpy
Next, we studied the robustness of the observed folding behavior against differences in hybridization strength, which could be caused by differences in GC content, temperature, or ionic conditions in experiments. To this end, we modified the hybridization potential between complementary beads to have the same cutoff distance but different enthalpies of hybridization of −6, −8, and −12 kcal/mol. Again, we simulated each of these cases ten times to obtain average kinetics. In the case of the lowest binding strength, we found that although staples are able to initiate onto the scaffold, they are unable to overcome the energy penalty of constraining the scaffold for additional binding; full binding of staples does not occur, and so full assembly does not occur during the simulation. In cases where the structure did fold, with −8 and −12 kcal/mol binding strengths, we again observed two different regimes of first-order kinetics as described above (Fig. 3b). The crystallinity and unrealized contacts behavior for these cases is consistent with that observed earlier for the parameterized hybridization enthalpy (Fig. 3c, d). In the −8 kcal/mol case, the two regimes appear to be present, but the transition is more subtle.

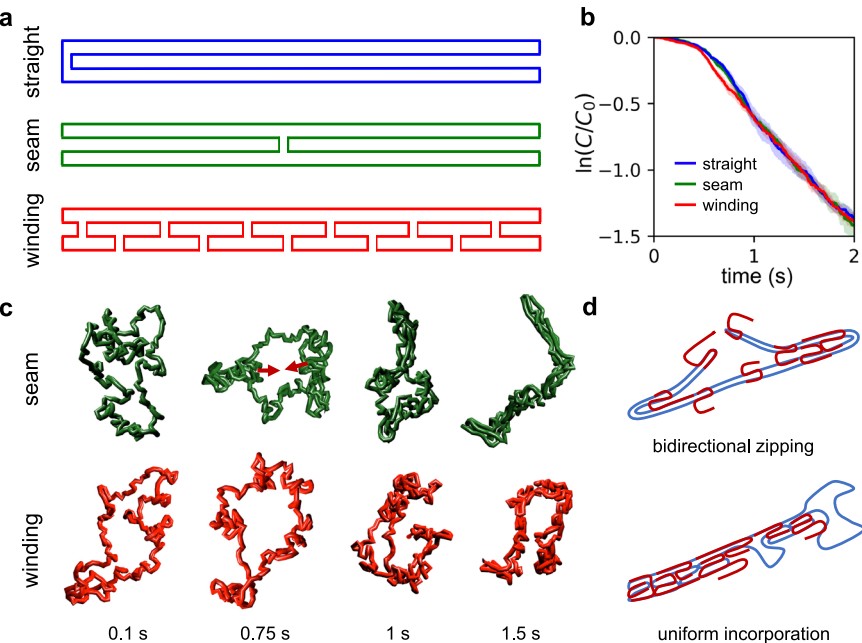

**Fig. 4 | Dependence of folding behavior on scaffold routing. a** Three different scaffold routings were employed for these simulations. **b** Kinetics of staple binding for the three designs. Note that straight and seam designs exhibit a change in the first-order rate constant, but the winding design does not. **c** Representative configurations of DNA scaffold captured from simulations highlighting differences in the assembly process for the two alternate scaffold designs. **d** Diagrams of the key folding mechanism for each design.

Generally, stronger binding was found to lead to faster kinetics at the beginning of folding because the rate of bridging and scaffold constraint is faster. This suggests that this portion of the folding process is reaction-limited for weaker binders (−6 to −10 kcal/mol): close encounters between a staple bead and its complementary scaffold bead are more likely to lead to completed hybridization when hybridization enthalpy is stronger due to a greater force pulling staple and scaffold together upon initiation of binding. For structures with high accessibility, once collapsed into the proto rod, origami folding becomes diffusion-limited (evidenced by similar kinetic rates in the second phase of folding for all hybridization strengths), whereas before collapse, there is some degree of reaction-limited behavior. These factors combine to result in faster kinetics for stronger binding enthalpies at the beginning but not the end of the folding process.

## Dependence on scaffold routing

An important factor in DNA origami design is scaffold routing. So far, its exact impact on folding kinetics and mechanism is not well understood. We thus created two additional scaffolds which, combined with the scaffold routing discussed above, represent the extremes of scaffold design (Fig. 4a): the previously introduced "straight" design only utilizes four crossovers at the ends and contains no crossovers in the middle. The second "seam" design contains a seam which is considered a common scaffold design. The final "winding" design has many crossovers, with one located every 16 bases along the entire scaffold. We simulated each of these designs and found that the seamed scaffold still exhibits zipping behavior like the straight scaffold routing; in this case, the zipping initiates from both ends rather than a single end, with this zipping action proceeding until both sections join at the center (Fig. 4c, d). Initiation occurs at both ends because two different bends in the scaffold are needed to form a proto rod compared to the straight structure which requires only one bend (Fig. 4b). The way the dynamics evolve in the seam structure prevents the seam from forming until near the end of the folding process, on average. This is because the formation of the seam requires all four duplex arms to be joined; it is thus much easier for the

system to first evolve into a pair of joined duplexes, then to "zip" toward the seam, bending from ¼ and ¾ of the way across the pair of joined duplexes.

The winding design does not exhibit zipping at all and instead folds at a constant kinetic rate throughout the entire folding process (following initial diffusion of staples), where it gradually becomes more structured until the final 4HB shape is attained (Fig. 4c, d). The correspondingly very low entropic penalty of loop constraint for each staple connection to the scaffold is likely responsible for the faster rate at the beginning of folding.

## Dependence on staple design

Another major factor in origami folding is staple design. To begin to explore the effect of staple design on the folding process, we created two additional 4HB designs with strategically placed long staples (retaining the original straight scaffold routing) that might serve to stabilize the structure at different locations once bound. The first modified design contains two 128-base staples at the center of the 4HB (but unmodified 32-nucleotide staples elsewhere), and the second modified design contains one 128-base staple at each end of the 4HB and unmodified 32-nucleotide staples in all other locations (Fig. 5a). Owing to the much larger size of these modified staples, we found that they associated much more slowly than the other, smaller staples, likely due to their slower rate of diffusion (Fig. 5c). This is consistent with the fact that all species are below their melting temperature, so the fastest-diffusing species should incorporate first. However, the kinetic curves indicate that all three designs are similar from a rate perspective beyond the first ~1 s (Fig. 5b). This is likely because the 4HB structure is highly accessible, and so the structure still folds in the same way even though the two large staples did not bind to and bridge the scaffold as quickly; the additional time that they take to incorporate into the scaffold does not have a significant effect on the overall kinetics, although it is clear from the binding time diagrams in Fig. 5c (see Methods) that the binding order changed upon the introduction of these staples. One notable behavior of the long center staples is that on the occasions where they do bind early in the folding process, they

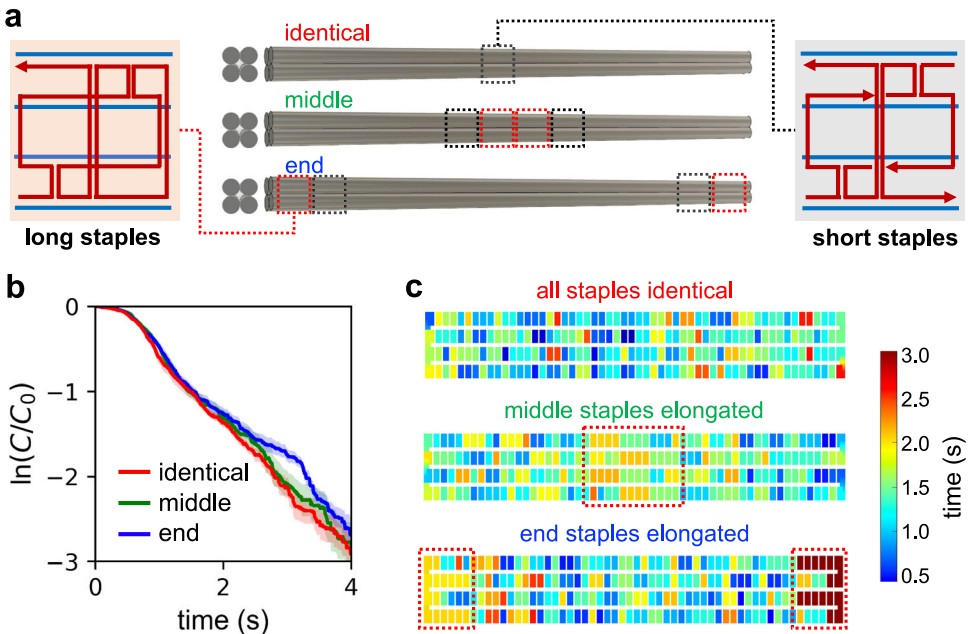

**Fig. 5 | Dependence of folding kinetics and binding order on staple design. a** Three different staple design layouts (all designs still 3584 nt/2.33 MDa). Modules containing long staples are shown in red. **b** Kinetics of folding. **c** Average binding time of each staple for each of the three staple designs.

automatically bridge the center of the 4HB and collapse the structure into a proto rod, irrespective of the folding progress, as opposed to the cooperative zipping observed in all other straight scaffold routing cases (see Supplementary Movies 1–9). This suppresses the two-phase folding behavior of this scaffold routing, hence the less apparent kink in the kinetic curve, while the end staples are unable to induce this premature collapse and therefore still produce the expected zipping effect. Overall, highly accessible structures seem to tolerate the addition of long staples during the folding process with a relatively minor effect on the folding mechanism, but this may have a much stronger effect when folding very large structures and when thermal annealing is used, since slow annealing effectively causes the binding order to correlate with the magnitude of free energy change of hybridization of each domain. This is because in the limit of slow thermal annealing (where the experimental timescale is much larger than the diffusion timescale of the staples), as temperature crosses the threshold at which a staple's incorporation into the DNA origami becomes thermodynamically favorable (the moment that $T < T_m$ of that staple), that staple should hypothetically be the next one to bind. This results in the order of staple incorporation being determined by the order of their melting temperatures.

### Folding of many-layer structures

Finally, we tested the ability of our model to capture the self-assembly of larger structures with significant interior geometry. We designed a low aspect ratio 32-helix bundle (32HB) and simulated its folding behavior (Fig. 6a). Interestingly, we found that the folding kinetics for this system are quite different from those of the more accessible 4HB structures discussed earlier. Instead of proceeding with a linear first-order rate, the kinetic curves obtained from all ten individual simulation runs are overall concave-up, in other words, on average, each addition slows the following staple addition (Fig. 6b). This is likely because as staples are added to the assembly, excluded volume accumulates around binding sites, reducing the accessibility of those binding sites to their complementary staples.

The folding of the 32HB begins with the scaffold in a random configuration. Just as in the case of the 4HB, persistent domains form, and the system must overcome an entropic barrier for these domains to coalesce and for the folding reaction to continue. However, instead

of collapsing rapidly into a near-final configuration as in the case of the 4HB, the unrealized intra-scaffold contact number requires much more time to reach its final value, and dwells in some cases. The 32HB has much more distributed folding character than the more accessible 4HB design. In Fig. 6d, e, we present the crystalline order and unrealized contacts of the 32HB from those ten simulations. We find that the spread of unrealized local contacts at 5 s is greater than 150, and some structures exhibit unfolded interiors, likely resulting from global topological defects forming when the scaffold is outside of the core, while others are very well formed (Fig. 6c, f and Supplementary Movies 10–11). Generally, the incorporation of more staples is correlated with a lower number of unrealized contacts and a higher structural crystallinity. However, faster coalescence into a proto form early on is not a perfect predictor of overall folding performance; based on the unrealized contacts metric, the best-folded structure at 1 s becomes the second worst performer at 3.6 s as it is caught in a global topological trap.

## Discussion

This work provides one of the first dynamic pictures of how DNA origami structures develop from their unfolded state into their final, crystalline form. This was achieved through the development of a switchable force field model which can capture all of DNA's relevant mechanical states and the transitions between them at a very coarse resolution of 8 nucleotides per bead, thereby extending the accessible timescale of dynamic simulations to the length of the folding process. Using this approach, we revealed in detail the hierarchical nature of DNA origami folding and connected this to folding kinetics. Furthermore, we investigated the folding of several different DNA origami designs and showed how folding behavior depends on common design parameters such as scaffold routing, staple design, and final nanostructure geometry.

One of the key results of this study was the discovery that some structures exhibit a global collapse phenomenon separating two different phases of kinetic behavior, where the structure transitions from an open, two-stranded character into a proto-rod via cooperative zipping. It is likely that this global collapse effect could be ubiquitous in DNA origami folding; indeed, intermediate folding products have been observed before[48]. Similar effects may have also taken place in

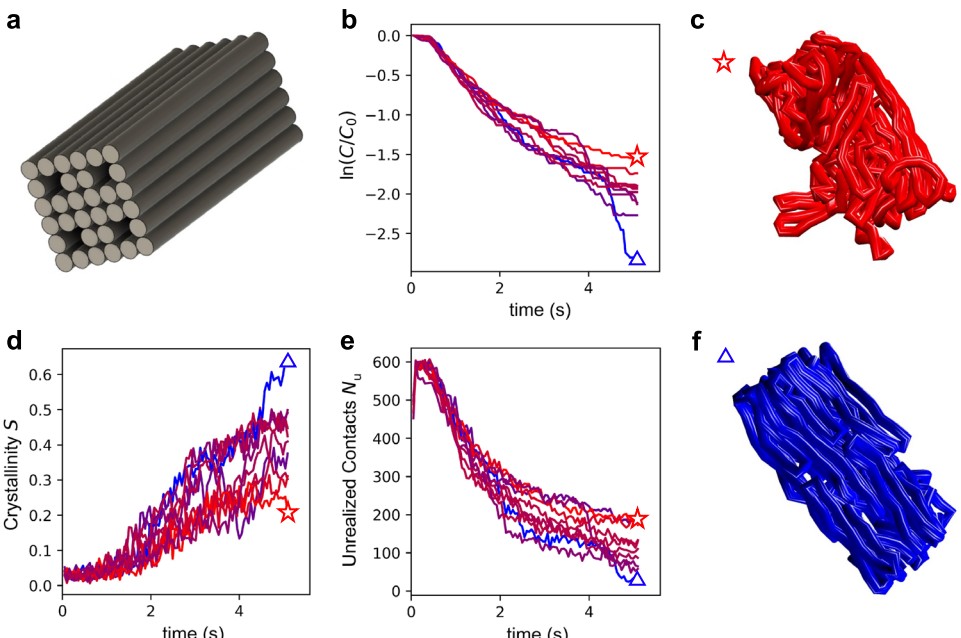

**Fig. 6 | Folding behavior of a 32HB structure with significant interior geometry.** **a** Design schematic of the structure (8192 nt/5.32 MDa). **b** Folding kinetics in terms of instantaneous staple concentration. Individual traces from ten independent folding simulation runs are colored according to the number of associated staples and are matched between panels **b**, **d**, and **e**. **c** Poorly folded 32HB with interior scaffold protruding. **d** Landau-de Gennes crystallinity parameter. **e** Number of unrealized contacts. **f** Well-folded 32HB with scaffold interior correctly located.

the study by ref. 28, where it was observed that the structure would dwell in a configurational state for some time before large portions of their bigger 42HB structure combined; it is possible that these larger sections were unable to combine until enough staples partially bound to them such that their combined enthalpy of hybridization could overcome the entropic penalty of combining the large chunks. Furthermore, this collapse effect may also explain their observation of multiphase kinetics, where they observed a double-exponential kinetic curve indicating multiple simultaneous kinetic rates of different events during folding. It is possible that, while these structures are quite different in size and geometry, they may still possess similar mechanisms of assembly. The persistence of large domains of DNA origami arrested during the folding process was also observed by ref. 22 in further support of these observations.

Another key result relates to the role of scaffold routing in folding kinetics, where we found that routings with many crossovers exhibited linear kinetics while the straight-across routing exhibited multiple kinetic regimes. Seamed structures were found to exhibit intermediate behavior. We hypothesize that while DNA origami folding may not always exhibit exactly one of the three reported behaviors, there likely exists some combination and superposition of each of these behaviors for larger structures. Regions of a large, complex structure with long straight sections of scaffold will likely fold with multiple kinetic regimes; sections of seamed structures will likely fold with intermediate kinetics, and sections with many crossovers may tend to fold with perfectly linear kinetics. This offers another potential explanation of the multiphase kinetics of folding observed by ref. 28.

For computational tractability, it has generally been assumed that origami folding is Markovian—a system containing no memory whose transition rates between states are determined based only on the current system state—when the system state is defined based purely on its hybridization state and configurations are ignored. This has formed the basis for multiple kinetic models of DNA origami folding[29] as well as tile assembly[49] and reconfiguration[50]. This may indeed be a good assumption when evaluating highly accessible structures that essentially reach structural equilibrium between each hybridization event

(as we also observe toward the end of the 4HB folding, where staples are added very slowly and the structure is essentially equilibrated during the post-collapse crystallization phase of assembly). However, using hybridization states to represent the absolute system state results in loss of information regarding the configurational state of the system which most likely has some effect on kinetics. This study has revealed that multilayer structures with significant interior geometry are likely to encounter global topological defects which severely slow or halt folding. This suggests that within any hybridization state may lay several configurational sub-ensembles, some of which do not have a viable path to correct folding or have escape times which may be very slow, potentially longer than the timescale of experiments. These sub-ensembles are likely energetically unfavorable compared to equilibrium states but are nonetheless likely to appear in the folding process. However, since kinetic models based solely on the hybridization state assume equilibrium at each state, these sub-ensembles are not sampled; kinetic models may thus significantly overpredict yield. On the other hand, if "folding momentum" (a sequence of folding events occurring out of equilibrium whose kinetics are significantly enhanced over the kinetics of transitions between equilibrated configurations) is at play[51], the kinetic model may underpredict yield owing to energetically unfavorable but kinetically favorable pathways that are not captured. Our dynamic simulations explicitly consider both the hybridization state of the system and its entire configurational state and, therefore should capture non-Markovian folding and folding momentum effects. Our model provides a tool to investigate these fundamental effects in more detail in the future as well as many other phenomena in DNA nanotechnology, such as competitive folding[19,52] and reconfiguration[53] of DNA origami structures.

From an experimental perspective, this study has provided multiple prospective folding mechanisms for various DNA origami structure designs. Several experimental techniques could be employed to test our predictions: firstly, small angle X-ray scattering (SAXS)[54] could potentially be used to determine the time-resolved radius of gyration of DNA nanostructures as they fold; this data could be directly compared to the easily computable radius of

gyration in simulations. Secondly, fluorescence experiments similar to those employed previously[28] could be used to evaluate the folding kinetics of the structures presented in this study. Also, the assembly process could be arrested periodically, and atomic force microscopy could be used to characterize the intermediate products, which could provide evidence of the cooperative zipping effect that we observed. Species tethered to surfaces and imaged using high-speed AFM during folding would provide a more high-throughput approach to this data gathering.

Our switchable force field strategy is quite general and could be relevant to any system that exhibits multimodal behavior—i.e., any process that possesses one behavior in one state and a second behavior in another state as well as transitions between those states—and whose modes are separated by a clear transition that can be described by the change of some scalar parameter. In the case of DNA nanotechnology, we have used this to capture the change in bending properties induced during the transition between ssDNA and dsDNA, and to enforce the square arrangement of nucleotides in double Holliday junctions. This concept could be applied to other systems like proteins, where a transition between peptides and their motifs possessing different mechanical behavior could be defined and realized at a coarser scale than is currently used in dynamic protein modeling.

While our model has provided many insights into DNA origami folding, it has the potential for further improvement. Some limitations will be addressed in the future, and some are inherent to the model's representation of DNA. Firstly, some design patterns in DNA origami are thought to result in the formation of local topological traps. For example, one of these patterns is the 14-7-14 staple crossover pattern, whereby staples with two 14-nucleotide domains on separate helices sandwiching a seven-nucleotide region with crossovers between them tend to incorporate defectively. The argument behind this is that if the two 14-nucleotide regions form first, the seven-nucleotide region will be topologically restricted from forming a correct helix due to the lack of a free end. Since our model does not contain the helical nature of hybridization, these types of folding defects cannot be captured with this model. In the future, this could potentially be addressed through the use of switchable excluded volume potentials to enforce a correct order of binding such that these topologically infeasible binding events cannot occur.

Secondly, there is a small energy (-1 $k_{B}T$) associated with base-stacking of adjacent staples that our model does not capture; this has recently been implicated in a nucleation barrier when assembling DNA origami with thermal annealing[30]. In the future, a potential will be added to address base-stacking between adjacent staples in order to capture this behavior. Third, this model does not currently capture a global twist arising from the assumptions used in square lattice DNA origami design. The effect of global twist on the self-assembly process is expected to be insignificant and thus was not included in this model. Fourth, this model currently only allows perfect complements to bind; in the future, we may incorporate a sequence-dependent potential between all possible pairs of beads to enable the misbinding of sufficiently complementary but imperfect scaffold-staple pairs. It is also of interest to use this model to study the dependence of folding behavior on thermal annealing used in origami fabrication, which would require additional development.

Our simulations were conducted at a high staple/scaffold concentration (1.6 μM) and at a 1:1 staple-to-scaffold ratio. While this is significantly higher than standard experimental fabrication conditions, using high concentrations is a typical accommodation required to tractably capture self-assembly, and we believe that the relevant phenomena will still be observable even at these concentrations. We also note that while it is atypical to fabricate DNA origami with an equal ratio of scaffold to staple strands, not including multiple scaffold or staple copies in the simulation eliminates most of the potential issues associated with ratios that are too small or too large, including multiple scaffold copies being bound together and multiple staple copies binding to a single scaffold. We thus believe the model represents folding at experimental (ideal) staple-to-scaffold ratios. In the future, we plan to probe the roles of concentration and staple-to-scaffold ratio in the folding mechanism.

Lastly, it is not certain that our simple proximity-based hybridization condition perfectly captures the hybridization process, nor does it perfectly address the intricacies of molecular and none-quilibrium physics. One could consider further extensions, such as adding a probabilistic component to the modeling of hybridization, akin to a Gillespie or kinetic Monte Carlo approach, to account for the fact that each bead collision will not necessarily lead to hybridization and the probability of incorporation will depend upon factors like orientation and momentum. This would provide a more realistic description of the process, but we do not expect these details to affect the overall folding mechanism presented here.

## Methods

### DNA origami design
The caDNAno1 and caDNAno2 design packages[55] were used for all DNA origami designs in this study. All designs are simplified representations of real designs, where all crossovers are located at exact multiples of eight nucleotides from each other. These designs can be found in Supplemental Figs. 11–16.

### Mesoscopic model
Every eight consecutive nucleotides in the system are modeled as a single representative bead, so the bead center represents the centroid of those eight nucleotides. The interactions between these beads are described by the following interaction potentials, whose parameters are summarized in Supplementary Table 1.

**Backbone potential.** We employ a simple harmonic potential for the DNA backbone:

$$U_{\text{stretch}} = \frac{1}{2}k_{\text{stretch}}(r - r_0)^2 \qquad (1)$$

where $k_{\text{stretch}}$ describes the spring potential of backbone stretching, $r$ is the separation distance between adjacent connected beads, and $r_0$ is the equilibrium value of this separation distance. To determine the equilibrium spacing $r_0$ between ssDNA beads, we simulated a 256 nt polyT strand using the oxDNA2 model ("oxDNA") and modified the mesoscopic backbone potential for ssDNA until it reproduced the distribution of ssDNA end-to-end distances. (Supplemental Fig. 7). For bound species (dsDNA and crossovers), we used the oxDNA simulations of the sheet structure that were also used for target representation validation to determine the distribution of separation distances between consecutive beads located on a continuous duplex scaffold and on crossovers and determined their appropriate $r_0$ values. We then iteratively modified our harmonic backbone potentials to reproduce these average separation values (Supplemental Fig. 8). Since the exact variance of consecutive backbone bead separation distances is not very important for capturing the global aspects of self-assembly, we make this potential soft to keep it robust to larger timesteps and do not over-emphasize the exact reproduction of the backbone separation distance distributions. However, we did establish very good agreement with the mean values.

**Excluded volume potential.** We use the short-range repulsive Weeks–Chandler–Andersen (WCA) potential[43] to capture the excluded

volume of DNA:

$$U_{\text{WCA}} = \begin{cases} 4\varepsilon\left[\left(\frac{\sigma}{r}\right)^{12} - \left(\frac{\sigma}{r}\right)^6\right] + \varepsilon & r \leq 2^{1/6}\sigma \\ 0 & r > 2^{1/6}\sigma \end{cases} \qquad (2)$$

where $\sigma$ and $\varepsilon$ represent size and energy parameters and $r$ is the separation distance between beads. The excluded volume potential acts between all non-complementary beads in the system (but not between complementary beads) and serves to prevent beads from coming unrealistically close to each other. The other goal of this excluded volume potential is to prevent strand pass-through. With fine models such as oxDNA, excluded volume parameters may be constructed in a way to simply prevent particle overlap, and since the particle size is larger than the spacing between beads, this automatically prevents the unphysical behavior of pairs of strands passing through each other. Our model does not naturally do this, so we made this potential stiffer to make this pass-through very unlikely. We thus selected an $\varepsilon$ value of 1 kcal/mol for our WCA potential to induce a 18 kcal/mol penalty to strand cross-through (Supplementary Fig. 5), which should make such events exceedingly rare near room temperature where these folding events are occurring.

**Hybridization potential.** Rather than having a specific sequence, the hybridization of beads is controlled by a pairing matrix that only allows beads with a fully complementary sequence ("complementary beads") to hybridize with each other with a binding potential. We assume that the sequence of the DNA origami and staples have been carefully chosen to reduce unintended staple binding. Hence, we do not consider the addition of hybridization potentials between non-complementary sequences to be necessary. When two complementary beads come within a specified distance of each other, they are pulled together by the hybridization potential and begin to share excluded volume. The hybridization potential takes the following form:

$$U_{\text{bind}}(r) = \begin{cases} \Delta U_{\text{bind}}\left(\frac{r - r_{\text{cut}}}{r_{\text{cut}}}\right) & r \leq r_{\text{cut}} \\ 0 & r > r_{\text{cut}} \end{cases} \qquad (3)$$

where $r$ is the distance between centers of the complementary beads, $\Delta U_{\text{bind}}$ is the binding energy change determined using umbrella sampling (see below), and $r_{\text{cut}}$ is the distance at which the beads become dehybridized. To parameterize the hybridization potential, we created two random complementary eight-nucleotide sequences with 50 percent GC content and conducted umbrella sampling of the unzipping process using their center of mass separation as a reaction coordinate with oxDNA at standard salt conditions (500 mM monovalent cation equivalent) and 300 K. We then used the Weighted Histogram Analysis Method (WHAM)[56] to obtain the potential of mean force (PMF) as a function of bead separation for the unzipping process. The PMF curve obtained from WHAM is understood to incorporate the natural tendency for beads to be further away from each other (translational entropy), $\rho(r) \propto 4\pi r^2$, where $\rho(r)$ is the probability density of finding two complementary beads at distance $r$; as well as the energy of hybridization, $\rho(r) \propto e^{-U(r)/k_{\text{B}}T}$, where $U(r)$ is the effective hybridization energy (including sterics and internal conformational entropy), $k_{\text{B}}$ is the Boltzmann constant, and $T$ is the system temperature. The probability density of finding two beads at distance $r$ is thus of the form $\rho(r) \propto 4\pi r^2 e^{-U(r)/k_{\text{B}}T}$. Applying Boltzmann inversion to this, we recover the free energy (PMF) $\Delta G_{\text{sep}}(r) = U(r) - 2k_{\text{B}}T\ln r + \alpha$ where $\alpha$ is a constant. For large values of $r$, $U(r)$ is zero and $\Delta G_{\text{sep}}$ takes the form $\Delta G_{\text{sep}}(r) = -2k_{\text{B}}T\ln r + \alpha$. Fitting this equation to the tail of our free energy landscape produces $\alpha$. Once $\alpha$ is known, we can subtract it and the entropic term from the PMF curve, yielding an

approximate curve representing $U(r)$, which we refer to as "hybridization enthalpy". This potential should produce the correct hybridization behavior (Supplemental Fig. 4). We say that this curve is approximate because our model is too coarse to capture the specifics of temperature-dependence of the enthalpic term for hybridization, and it is also too coarse to capture some vibrational entropy terms. Ultimately, we obtain a cutoff distance of 2 nm with approximately −10 kcal/mol enthalpy of hybridization holding the strands together at 300 K, with a small barrier to hybridization. Since this model is intended to be quite coarse, we further simplify the potential to be linear up to 2 nm and cutoff thereafter with no hybridization barrier.

**Bending potential.** The bending properties (persistence length) of DNA are captured using a simple bending potential[57]:

$$U_{\text{bend}} = \frac{1}{2}k_{\text{bend}}(\theta - \theta_0)^2 \qquad (4)$$

where $\theta_0$ is the equilibrium bending angle, set to zero for continuous duplex and $\pi/2$ for crossover sections, $k_{\text{bend}} = k_{\text{B}}Tl_{\text{p}}/r_0$ is the bending constant which depends on the persistence length $l_{\text{p}}$ of dsDNA. ssDNA can be modeled as a freely jointed chain with excluded volume interactions between nonconsecutive beads since our bead size will be nearly the Kuhn length of ssDNA[58,59]; hence, the bending potential is not applied to ssDNA. When three or more consecutive scaffold beads become bound, bending force fields are activated to indicate the transition to duplex DNA. The bending potential only acts on the DNA scaffold in order to correctly capture the 50 nm persistence length[60] and allows unhybridized scaffold beads to assume their normal persistence length of ~1–2 nm[58,59].

**Crossover stabilization potential.** Crossovers behave fundamentally differently from single- and double-stranded DNA in that they exhibit an energetic minimum consisting of three or more strands combined together; the geometry of DNA and base-stacking interactions confer the characteristic Holliday junction shape. One important aspect to consider is that these Holliday junctions do not form a perfect planar configuration but rather assume a handed orientation. As Snodin et al. observed[42], oxDNA is unable to capture the handedness of Holliday junctions observed in cryo-EM data. However, we argue that this is of little importance since in DNA nanotechnology, these crossovers are usually confined to a planar configuration by opposing Holliday junctions located a few turns away. Rather than attempt to recreate a handedness in the Holliday junction, we elect to constrain crossovers into a planar configuration. To constrain junctions, we apply a harmonic constraint like that used for backbone constraints to the particle whose 5′ end enters the junction and to the particle following the junction at a distance of $r_{\text{eq}} = \sqrt{r_{\text{axial}}^2 + r_{\text{radial}}^2}$, where $r_{\text{axial}}$ is the continuous duplex value of 2.725 nm and $r_{\text{radial}}$ is the crossover distance value of 2.4 nm. We additionally apply this in the reverse direction to the particle whose 3′ end enters the junction.

$$U(r)_{\text{cross-stretch}} = \frac{1}{2}k_{\text{stretch}}(r_{i,i+2} - r_{\text{eq}})^2 \qquad (5)$$

We also apply a 90-degree switchable bending angle constraint to crossovers instead of the usual zero-degree angle for continuous duplex. This potential once again only applies to the DNA scaffold (and not staples) as in the case of the linear bending potential.

### Switchable force field Brownian dynamics simulations
We used custom-developed switchable force field simulation software, which we call DNAfold, for all folding simulations in this study. This software employs the potentials described earlier and simulates

the motion of staple and scaffold beads assuming overdamped Langevin dynamics (Brownian dynamics). Complementarity between staple and scaffold beads is described by pairing matrices which enable hybridization to occur based on the hybridization potential described above. The software performs simulations as follows: the program first executes an importer function which reads a standard caDNAno design file, produces the requisite pairing matrices, and identifies which beads are connected as staples and scaffold. The software then places all species in a simulation box at random positions (staples and scaffold are placed such that their bond distances are at their equilibrium values). Staples and scaffold are simulated at a 1:1 ratio, and no duplicates are present in the simulation box to alleviate concerns about misfolding related to multiple identical scaffolds binding to a single staple or multiple identical staples binding to a single scaffold. The box begins larger than the final box size to prevent overlaps and then shrinks to its final size over the first portion of the simulation. The simulation box is periodic, and the minimum image convention is employed when calculating forces unless the periodic box is manually deactivated, in which case forces are calculated based on absolute position. With each timestep, the software calculates all forces between all beads based on the potentials described in the text and applies statistically valid Gaussian-distributed stochastic forces. A second-order Runge–Kutta algorithm[61] is used to integrate the equations of motion. This is repeated for the user-specified number of timesteps until the end of the simulation. The software periodically records the coordinates of all beads into a trajectory file and records the number of bound species, which is used to compute kinetic curves. The software also records each incidence of hybridization or dehybridization and the species involved, which can be used to determine the first binding time, on and off rates, and other relevant properties. The software assumes that each eight-nucleotide particle has a hydrodynamic diameter of 2.7 nm based on the average pitch between beads. The real timestep used in simulations is 5 ps. Stokes' law is used to determine the drag coefficient, $\gamma = 6\pi\eta R$, where $\eta \equiv \eta(T)$ is the temperature-dependent viscosity of water; $T$ is the temperature of the fluid; and $R$ is the hydrodynamic radius of the species undergoing Brownian motion. We do not consider explicit interparticle hydrodynamic interactions but rather consider the isotropic diffusion tensor for computational efficiency. DNAfold is written in C++ (see Code availability).

### Principal component analysis (PCA)

This technique extracts essential motions of molecules by computing the combinations of particle motions in a molecular trajectory which result in the largest deviations from the mean structure of that body of beads in the simulation[62]. PCA was conducted on three systems: the nucleotide coordinates from oxDNA simulations, coarsened coordinates of the same oxDNA simulations to match our model (centroids of every eight nucleotides), and, finally, our own mesoscopic model's bead coordinates in BD simulations. A custom PCA code was written for this study and can be found on GitHub at https://github.com/marcello-deluca.

### Kinetics of DNA origami self-assembly

In this study, each non-hybridized eight-nucleotide staple domain is considered to be an independent species. The concentration of free staple domains inside the simulation box is denoted by $C$. The initial concentration of free staple domains inside the simulation box $C_0$ is simply the concentration of staple domains in the simulation since the folding simulations begin with no hybridized species. All folding simulations were conducted at a final box size of 100 nm by 100 nm by 100 nm, corresponding to 1.6 μM staple and scaffold concentration. Assuming first-order kinetics, plotting $-\ln(C/C_0)$ should yield a linear curve whose slope is the first-order rate constant.

We acknowledge that the observed kinetics ignore the points raised by Ouldridge regarding local concentration fluctuations[63] and note that a transformation may be necessary to better compare this data to real experiments. Design files for all structures can be found in the supplement.

### Crystallinity order parameter

The Landau-de Gennes order parameter describes the overall crystallinity of structures by observing the ordering of the vectors pointing to their neighbors in a straight line: $Q_{\alpha\beta_i} = \langle \frac{3}{2}\hat{t}_{i\alpha}\hat{t}_{i\beta} - \frac{1}{2}\delta_{\alpha\beta} \rangle$ where $\hat{t}$ is a tangent vector, $i$ is the particle number, $\alpha$ and $\beta$ are Cartesian dimensions (there should be three, for a total of nine entries in the Q matrix), and $\delta_{\alpha\beta}$ is the Kronecker delta evaluating to 1 when $\alpha = \beta$ and 0 otherwise. Averaging over all beads and computing the maximum eigenvalue of the resulting averaged $\mathbf{Q}$ matrix provides an instantaneous value $S = \lambda_{\max}(\mathbf{Q})$.

### Intra-scaffold contact parameter

This quantity describes the instantaneous number of unrealized intra-scaffold contacts in a DNA origami design:

$$N_u = \sum_i \sum_{j \in X_i} \begin{cases} 1 & \left| \|\Delta\mathbf{r}_{i,j}\| - \|\Delta\mathbf{r}_{i,j}^I\| \right| > 3\,\text{nm} \\ 0 & else \end{cases} \quad (6)$$

where $X_i$ is the set of all intra-scaffold beads located less than 2.5 nm from each other in the idealized configuration and not directly connected via the scaffold backbone, $\Delta\mathbf{r}_{i,j}$ is the Cartesian distance between beads $i$ and $j$, and $\Delta\mathbf{r}_{i,j}^I$ is the Cartesian distance between those beads in the idealized, folded configuration. This parameter describes how well-aligned nonbonded scaffold neighbors (neighbors that are separated in space by no more than one bead diameter in the fully assembled structure and are also not directly attached via the backbone) have become in the global structure. We consider this parameter to be a good measure of local order in assembling DNA origami structures.

### First binding time calculations

First binding times were calculated based on the first recorded instance of a bead binding to the scaffold. The mean first binding times reported in this study were averaged over all ten simulations of each system studied.

### Weighted histogram analysis method

The WHAM program provided in ref. 56 was used for inferring free energy profiles from umbrella sampling simulations when computing the hybridization potential in this study.

## Data availability

The data generated in this study have been deposited in Github at https://github.com/marcello-deluca/dnafold without any restrictions.

## Code availability

The codes used in this study have been deposited in public GitHub repositories (https://github.com/marcello-deluca/dnafold-analysis and https://github.com/marcello-deluca/dnafold) without any restrictions[64,65].

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

## Acknowledgements

We would like to thank Qinyi Lu for contributing 4HB designs from which the ones used in this study were derived. We would also like to thank Fabian Schneider for helpful discussions. This work is supported by the National Science Foundation (Grant Nos. CMMI-1921955 and EFMA-1933344), and the US Department of Energy (Grant No. DE-SC0020996). M.D. is supported by the National Science Foundation Graduate Research Fellowship (Grant No. 2139754). We acknowledge resources from NSF ACCESS and the Duke Computing Cluster for carrying out oxDNA and BD simulations.

## Author contributions

M.D. and G.A. conceived the approach. M.D. developed the simulation software, conducted simulations, and performed all analysis. D.D. assisted with code development, model validation, and kinetic time scaling. M.D. and G.A. wrote and edited the manuscript. G.A. provided advice and guidance on simulation techniques and analysis. Y.K., T.Y., M.P., and C.C. provided advice on different DO design considerations to investigate and provided insight into simulation results. All authors commented on the manuscript.

## Competing interests

The authors declare no competing interests.
