## [Peer Review File · Nature Communications]

Mechanism of DNA origami folding elucidated by mesoscopic simulationsREVIEWER COMMENTS

Reviewer #1 (Remarks to the Author):

The manuscript presents a novel coarse-grained model to study the mechanism of folding of DNA origami nanostructures.

Though extremely simplified:

- this model does not account for the fact that in experiments staple structures hardly ever have domains of same binding strength

- mis binding, unbinding and strand displacement are considered to be part of the folding process of a non-idealized origami and are excluded from this model,

this model does shed light on the implication of scaffold and staple design under the proposed conditions. An interesting result is the fundamentally different incorporation behavior of staples in the case of the 32HB structure vs the more open 4HB design.

I would suggest publishing the manuscript after addressing some minor issues.

1) I would propose for the investigated design files to be added as json files to the manuscript supplement / GitHub or to nanobase.org

2) It was really frustrating to not be able to test the software.

The provided github link to the main simulation package has not enough documentation to compile the package, the analysis link <https://github.com/marcello-deluca/dnafold-analysis> is probably leading to a closed access repository and is thus not accessible.

Along these lines references 64 and 65 lead towards the same repository and p.31 PCA code link is not available.

3) Methods p.30 "The box begins larger than the final box size to prevent overlaps and then shrinks to its final size over the first portion of the simulation." Please elaborate more on this relaxation process, what is the stop condition?

to which experimental concentration of staples and scaffold would this correspond to?

Reviewer #2 (Remarks to the Author):

This study aims to elucidate the dynamics of DNA origami folding using a switchable force field that simulates all relevant mechanical states and the transitions between them at a coarse resolution (8 nucleotides per bead).

The model allows the simulation of the entire folding process, examining the rate and the hierarchy of individual identifiable steps. The researchers carefully investigated the folding of different DNA origami designs (mostly rods), focusing on how different parameters like scaffold routing, staple design, and final nanostructure geometry can impact the folding behaviour. A key finding was the occurrence of a global collapse phenomenon in some structures, involving phase transitions between open and proto-rod states. The authors propose that this phenomenon might explain previously observed multi-phase kinetics.

This work is of importance for the advancement of the DNA origami field (and not only) because it proposes a novel method to analyse and visualise the dynamics of structures' folding. The method proposed by the authors provides a rationalization of many experimental observations in the field and the improvements proposed by the authors in the discussion section promise to be useful in revealing some aspects that have not yet been understood. The work is also well written and clear in most parts and very pleasant to read.

I am therefore favorable to publication of this work in Nature Communication, upon the issues below have been addressed.

(1) In figure 1, I would indicate that 1 bead is 8 nt for a square-lattice design.

(2) ..'if three or more consecutive beads of DNA scaffold become bound ... to three or more staple beads, a bending potential is activated to enforce the persistence length of dsDNA in only this region.'

Why do the authors say 'three OR more'. Does it mean a minimum of three?

If I understand correctly, this would mean that the model can capture the hybridisation of at least $8 \times 3 = 24$ consecutive nucleotides?

(3) 'Crossovers in DNA origami design also exhibit switchable behavior with their bending potentials being activated when enough portions of the crossover become bound..'

can the authors please specify what do they consider as 'enough portions'?

(4) "the seamed scaffold still exhibits zipping behavior like the straight scaffold routing; in this case, the zipping initiates from both ends rather than a single end, with this zipping action proceeding until both sections join at the center"

I find difficult to explain this progression of folding. After the initial folding from the extremities of the four "duplex arms", I would have expected to see two zipping events, each one involving half of the 4HB and proceeding from the bend (now the seam) to the edges, as in the straight routing. Can the authors elaborate more on this point? Please also see my comment below at point (10).

(5) "The kinetics are slightly faster for the seam design at the beginning of the simulation because this structure adopts a more open configuration when it begins to zip shut than the structure with a straight scaffold routing and thus can fold more quickly".

This explanation is not fully clear to me. The first phase of the process, when the structure adopts a more open conformation, takes place before the zipping. Why zipping should affect it?

(6) the folding of the winding design was explained by the very low entropic penalty of loop constraint for each staple connection to the scaffold. What can the authors say about the bending potential here? is the winding design as easy to bend as the straight one?

(7) In figure 5e, please indicate which staples are the long ones.

(8) “Owing to the much larger size of these modified staples, we found that they associated much more slowly than the other, smaller staples, likely due to their slower rate of diffusion.”

Please, refer here to Figure 5b (first phase < 1s).

(9) “while the end staples are unable to induce this collapse and therefore still produce the collapse effect” Probably: “...still produce the zipping effect”

(10) I very much appreciate the discussion part and the proposal of possible improvements for the future.

Something more could be said about the role of hybridisation enthalpy on the folding progression. Indeed, previous studies made on seam-containing structures (doi.org/10.1021/acsnano.5b05972) have shown that differences in the T_m of the staples and edges affect the region of the structure that starts to fold or unfold.

This concept is also shortly mentioned in the results section when the authors state that “... slow annealing effectively causes the binding order to correlate with the magnitude of free energy change of hybridization of each domain.”

Can the authors elaborate more on this point?

(11) along this reasoning: do the authors think that their approach might be improved in the future to simulate the hysteresis often observed in folding/unfolding experiments?

(12) could this method help to explain the folding of some origami sheets into two distinct but well-formed isomers

(<https://doi.org/10.1002/anie.201402973>; doi.org/10.1038/s41467-019-09002-6) or their reconfiguration upon addition of triggers? ([10.1126/science.aan3377](https://doi.org/10.1126/science.aan3377)).

sequence-dependence, hysteresis and isomerization are important phenomena in protein

folding and reconfiguration. Hence, I imagine that using this model to simulate the folding of some representative protein candidates could be very instructive and also help revealing other aspects of DNA origami folding in general.

(13) a last point: can the authors clearly state the size and MW of the structures they have simulated? which is the range of sizes and MW that can be reasonably simulated by their approach?

Reviewer #3 (Remarks to the Author):

DeLuca et al describe a coarse-grain model that provides access to information about the assembly of DNA origami structures. Interaction between staple and template is modeled at a domain level where each domain represents 8 nucleotides. The model uses a switchable force field to capture the mechanical properties of single-stranded and double-stranded domains. The model is first benchmarked against oxDNA which has been used extensively to characterise folded DNA origami structures. It is then used to explore isothermal assembly pathways of (small) origami systems.

The model is a welcome addition to the tools available for modelling DNA origami assembly and potentially allows end users to begin to design not just final assembled structure but assembly pathway too. That said, its adoption will depend upon ease of use (note the value of oxDNA.org and oxView in opening access to new users by lowering the barrier to adoption).

Some features of the model are not obvious to me:

- do simulations contain more than one copy of each staple? Staple excess is an experimentally important though poorly understood parameter and one that models tend not to capture well (staple excess seems to facilitate folding in experiment and trap misfolds in simulation - likely due to effectively irreversible interactions at simulation temperature).

- are staple binding events readily reversible at the simulation temperature?

- what effect does the temperature have on assembly, how was the simulation temperature decided, how close is it to the melting temperature of the structures under study?

Folding of a 4-helix bundle is described as a two stage process. Half of the staple contacts form relatively quickly forming an extended structure with staples linking distant domains forming a loose 2-helix bundle. The structure is then folded in half and zippers shut from one end. The remaining contacts form more slowly in this stage, this is attributed to excluded volume effect. The transition between the two stages is not characterised in detail. The scaffold routing is shown to influence the zipping behaviour, this, for me, is the clearest conclusion of the work and one which the origami designer has most control over.

I'm not sure that I understand what is going on during the two stage assembly. It is odd that the average staple behaviour is discussed when information about individual staples is available (e.g. as presented in Fig 5C). It would be worth investing time into how to present these results. I would like to be able to better understand the order in which contacts are made (for example contact maps colored by domain incorporation time).

Something has gone wrong with the plots in 5C. It would help to have a little white space between domains, to make sure that domains line up in columns (perhaps that is what was intended - it looks a little like two such plots might have been overlaid)

Message To All Reviewers:

We would first like to thank the reviewers for taking the time to review our work and for suggesting valuable changes to the manuscript. During revisions, we uncovered a small bug in our code due to which our simulated model was not reproducing the 50 nm persistence length of dsDNA. We have since corrected this bug and re-run all of the simulations presented in the manuscript. We additionally rescaled the timescale of our simulations based on the hybridization of DNA, adjusting the time to best match the hybridization kinetics to literature values. This is shown in a new supplementary figure (Fig. S18). Despite these changes, the findings / conclusions of the manuscript remain unaffected, though, as expected, the precise values of various plotted quantities are slightly different in the revision:

- The histograms in Figure 2 are now a much better match due to the correctly represented persistence length.
- Figures 3-6 have different quantitative values for their charts but the general trends and behaviors remain unchanged.
- The end-to-end distance distributions in the supplement for 256bp dsDNA now match much more accurately (Fig. S6).
- In the investigation of the role of scaffold routing on the folding process, the seam structure now initially folds at a statistically indistinguishable rate to the straight structure (Fig. 4). The key finding that the winding structure still does not exhibit structural collapse / zipping and a subsequent change in the folding rate is preserved, consistent with the original findings.

Response to Reviewer 1:

The manuscript presents a novel coarse-grained model to study the mechanism of folding of DNA origami nanostructures. Though extremely simplified [...] this model does shed light on the implication of scaffold and staple design under the proposed conditions. An interesting result is the fundamentally different incorporation behavior of staples in the case of the 32HB structure vs the more open 4HB design.

We thank the reviewer for their positive feedback and for helping us to improve our manuscript.

This model does not account for the fact that in experiments staple structures hardly ever have domains of same binding strength.

This model does account for stronger binding domains by way of *domain length*, having multiple consecutive staple beads in a single domain when, for example, there are 16 nucleotides between crossovers instead of 8. At the same time, we do acknowledge that the model does not capture the (more minor) but still important differences between *sequence-specific* binding strengths. **This has been clarified in the text on page 8 of the revised manuscript.**

In addition, we admit that the use of exclusively multiples of 8 nucleotides for domains is a limitation of the model (dictated by the square lattice packing of dsDNA helices) and would require additional treatment and development to implement other packing geometries (e.g., honeycomb lattice) in future versions of this model / software.

Misbinding, unbinding and strand displacement are considered to be part of the folding process of a non-idealized origami and are excluded from this model.

The reviewer is indeed correct that misbinding and strand displacement are excluded from this model. However, unbinding is captured by this model (see the following passage “*This binding is reversible, so pathways requiring misbound species to dissociate for folding to complete can be properly represented.*” on page 8 explaining this). **We have included additional clarification on this point on page 8.**

To capture the former two, one would need to implement a sequence-dependent parameterization of the hybridization potential which will require extensive free energy calculations and has been planned as a separate project. In addition, misbinding and subsequent strand displacement by the correct species is to some extent avoided by careful sequence selection when designing staples. We thus believe that this is a good first approximation to make in our model, especially because of the reduction in folding timescale achieved by not considering misbinding.

We agree that the origami is in some sense idealized; however, we believe that this idealized representation still captures the most essential physics of the folding process and provides valuable information about the overall mechanism of real DNA origami folding.

I would propose for the investigated design files to be added as json files to the manuscript supplement / GitHub or to nanobase.org

Thank you for your suggestion. The JSON files have been added to a folder titled “DNA origami design files” in the revised manuscript supplement.

It was really frustrating to not be able to test the software. The provided github link to the main simulation package has not enough documentation to compile the package, the analysis link is probably leading to a closed access repository and is thus not accessible. Along these lines references 64 and 65 lead towards the same repository and p.31 PCA code link is not available.

We sincerely apologize for these problems. We have now corrected the issue with the analysis tools repository, made significant modifications to the source code to make it more user friendly, and included

a user guide with example folders in the GitHub repository (the user guide is hosted separately at <https://daniel-duke.github.io/dnafold-docs/>). These usability improvements include:

- Modifying the parameter ingest to take an input file in .txt format with parameters specified in key=value format so that the software only needs to be compiled once (we have included documentation and examples to make it easy to formulate these input files).
- Reducing the number of arguments required to run a simulation to two: the executable and the input, e.g. `./dnafold input.txt`.
- We have prepared a detailed user manual to increase the usability of the code. The manual contains three sections: (1) installation and running instructions; (2) descriptions of all the parameters, their units, and their defaults; and (3) a description of how to create compatible caDNAno designs. The last section contains several examples of designs that DNafold would successfully simulate, as well as designs that would throw flags or produce undefined behavior.
- We have improved the JSON ingest behavior to automatically format the topology file so that the topology and trajectory can be opened using the Open Visualization Tools (OVITO) package or any package that reads LAMMPS formatted trajectories, thereby allowing the visualization all of the individual DNA strands and their bonds.
- Simulations now generate a simulation metadata file that documents the value of every user-set parameter in the simulation.

We now feel confident that a user will be able to independently run DNafold to simulate their own DNA origami designs with parameters of their own specification. We note that we are continuing to improve this software and intend to expand the available features and sophistication of the model. Additionally, we request that users compile and run this software on MacOS, Windows Subsystem for Linux, or Linux operating systems. Building the software in a native Windows environment is not currently supported.

Methods p.30 "The box begins larger than the final box size to prevent overlaps and then shrinks to its final size over the first portion of the simulation." Please elaborate more on this relaxation process, what is the stop condition?

The box size begins at size *box_size* which is a variable set in the program, in nanometers. A variable in the program called *shrink_rate* determines by how much the box should be shrunk at each step. Another variable *final_box_size_ratio* determines the final size relative to the initial size. This value is determined based on what concentration the user desires; The box stops shrinking once it is of size *final_box_size_ratio* x *box_size* or smaller. These values can all be set in the simulation input file and are then parsed to set these variables in the program, **which is now well explained in the user manual.**

To which experimental concentration of staples and scaffold would this correspond to?

The final box size for the 4HB and 32HB simulations is 100 nm x 100 nm x 100 nm which corresponds to a staple and scaffold concentration of 1.6 μ M. While this is significantly higher than standard fabrication conditions, using high concentrations is a typical accommodation required to tractably capture self-assembly, and we believe that the relevant phenomena will still be observable even at these concentrations. We also note that while it is atypical to fabricate DNA origami with an equal ratio of scaffold to staple strands, not including multiple scaffold or staple copies in the simulation eliminates most of the potential issues associated with staple-scaffold ratios that are too small or too large, including multiple scaffold copies being bound together and multiple staple copies binding to a single scaffold. We thus believe the model is representing self-assembly behavior at experimental/ideal staple-scaffold ratios. **We have explained this point on pages 24, 25, and 32 of the manuscript.**

Response to Reviewer 2:

This study aims to elucidate the dynamics of DNA origami folding using a switchable force field that simulates all relevant mechanical states and the transitions between them at a coarse resolution (8 nucleotides per bead).

The model allows the simulation of the entire folding process, examining the rate and the hierarchy of individual identifiable steps. The researchers carefully investigated the folding of different DNA origami designs (mostly rods), focusing on how different parameters like scaffold routing, staple design, and final nanostructure geometry can impact the folding behaviour. A key finding was the occurrence of a global collapse phenomenon in some structures, involving phase transitions between open and proto-rod states. The authors propose that this phenomenon might explain previously observed multi-phase kinetics.

This work is of importance for the advancement of the DNA origami field (and not only) because it proposes a novel method to analyse and visualise the dynamics of structures' folding. The method proposed by the authors provides a rationalization of many experimental observations in the field and the improvements proposed by the authors in the discussion section promise to be useful in revealing some aspects that have not yet been understood. The work is also well written and clear in most parts and very pleasant to read.

I am therefore favorable to publication of this work in Nature Communication, upon the issues below have been addressed.

We thank the reviewer for their positive feedback and for helping us to improve our manuscript.

In figure 1, I would indicate that 1 bead is 8 nt for a square-lattice design.

Thank you for your comment – **we have made the requested change to Figure 1.**

“if three or more consecutive beads of DNA scaffold become bound ... to three or more staple beads, a bending potential is activated to enforce the persistence length of dsDNA in only this region.” Why do the authors say “three OR more”. Does it mean a minimum of three? If I understand correctly, this would mean that the model can capture the hybridization of at least $8 \times 3 = 24$ consecutive nucleotides?

To clarify, the model can capture the hybridization of any multiple of 8 nucleotides, even a single bead. However, to enforce a bending potential, a minimum of three representative particles are required. This is because in order to determine bending forces on a set of particles, one must compute the angle between interparticle vectors in order to determine the alignment force used to enforce the correct persistence length. The model can thus only capture the change in bending persistence length of DNA strands upon hybridization of 24 or more consecutive nucleotides. We do not believe that this constraint is a limitation because the bond length between consecutive beads (8 nucleotides, ~ 2.75 nm) is much smaller than the persistence length of double stranded DNA (~ 50 nm). **This is clarified on page 8 of the manuscript.**

Referencing the passage “Crossovers in DNA origami design also exhibit switchable behavior with their bending potentials being activated when enough portions of the crossover become bound,” can the authors please specify what do they consider as “enough portions”?

We thank the reviewer for raising this point. By ‘enough portions’, we mean three consecutive connected beads of which at least two constitute the crossover. **We have clarified this on page 9 of the manuscript.**

Referencing the passage “the seamed scaffold still exhibits zipping behavior like the straight scaffold routing; in this case, the zipping initiates from both ends rather than a single end, with this zipping action proceeding until both sections join at the center,” I find it difficult to explain this progression of

folding. After the initial folding from the extremities of the four “duplex arms”, I would have expected to see two zipping events, each one involving half of the 4HB and proceeding from the bend (now the seam) to the edges, as in the straight routing.

Since the two folds occur indiscriminately, on average, they both occur at the same time and so average kinetics indicate a single transition in the rate. We note that we also expected the fold to initiate from the seam; however, the way the dynamics evolve in this system prevents the seam from fully forming until near the end of the folding process, on average. This is because the formation of the seam requires all four “duplex arms” to be joined. It is thus much easier for the system to first evolve into a pair of joined duplexes, then to “zip” toward the seam, bending from $\frac{1}{4}$ and $\frac{3}{4}$ of the way across the pair of joined duplexes (see figure 4). **We have modified the manuscript in page 16 to better explain this nuanced behavior.**

Referencing the passage “The kinetics are slightly faster for the seam design at the beginning of the simulation because this structure adopts a more open configuration when it begins to zip shut than the structure with a straight scaffold routing and thus can fold more quickly,” This explanation is not fully clear to me. The first phase of the process, when the structure adopts a more open conformation, takes place before the zipping. Why should zipping affect it?

Thank you for your question. After modifying the bending potential to more accurately reflect the 50 nm persistence length of double-stranded DNA, we witnessed that this effect was diminished. We have removed it from the manuscript.

The folding of the winding design was explained by the very low entropic penalty of loop constraint for each staple connection to the scaffold. What can the authors say about the bending potential here? is the winding design as easy to bend as the straight one?

Thank you for your question. Our interpretation of this question is that the reviewer is wondering whether the bending stiffness of the overall structure is greater for the straight scaffold or winding scaffold design. While investigating the overall bending stiffness of folded structures was not one of the main aims of this study, we speculate that the presence of many additional scaffold crossovers may affect the overall bending stiffness of this structure. This is a good point and an important design consideration that we could investigate in the future.

In figure 5e, please indicate which staples are the long ones.

Thank you for your suggestion. **We have now indicated which staples are long in Figure 5a and c.**

Referencing the passage “Owing to the much larger size of these modified staples, we found that they associated much more slowly than the other, smaller staples, likely due to their slower rate of diffusion.” Please, refer here to Figure 5b (first phase < 1s).

Thank you for your suggestion. We believe the reviewer is referring to the average incorporation time plot in figure 5c, which demonstrates that the longer staples bind later in the process. Figure 5b only shows *average* staple incorporation but cannot differentiate between long and short staples. **We have thus referenced figure 5c in the text in the revised manuscript.**

Referencing the passage “while the end staples are unable to induce this collapse and therefore still produce the collapse effect,” this should probably be modified to “...still produce the zipping effect”

After re-running these simulations with the correct persistence length, this effect was diminished; **we have thus removed this passage from the manuscript.**

Something more could be said about the role of hybridisation enthalpy on the folding progression. Indeed, previous studies made on seam-containing structures (doi.org/10.1021/acsnano.5b05972) have shown that differences in the T_m of the staples and edges affect the region of the structure that starts to fold or unfold. This concept is also shortly mentioned in the results section when the authors state that "... slow annealing effectively causes the binding order to correlate with the magnitude of free energy change of hybridization of each domain." Can the authors elaborate more on this point?

The free energy change of hybridization dictates the thermodynamics of staple incorporation and so in the limit of slow thermal annealing (where the experimental timescale is much larger than the diffusion timescale of the staples), as temperature crosses the threshold at which a staple's incorporation into the DNA origami becomes thermodynamically favorable (the moment that $T < T_m$ of that staple), that staple should hypothetically be the next one to bind. This results in the order of staple incorporation being determined by the order of melting temperature. **We have mentioned this point more clearly in the revised manuscript on page 18.**

Along this reasoning: do the authors think that their approach might be improved in the future to simulate the hysteresis often observed in folding/unfolding experiments?

If the reviewer is alluding to the specific cooperative effects (such as staple-staple base stacking) that are hypothesized to cause folding/unfolding thermal hysteresis, we plan to incorporate base stacking effects in a future version of this model which should enable us to observe this effect. On the other hand, if the reviewer meant this point more generally, *i.e.*, folding and unfolding using temperature in general, we also have planned to incorporate temperature-dependent hybridization potentials into our model which will more accurately capture melting and refolding.

Could this method help to explain the folding of some origami sheets into two distinct but well-formed isomers (<https://doi.org/10.1002/anie.201402973>; doi.org/10.1038/s41467-019-09002-6) or their reconfiguration upon addition of triggers? ([10.1126/science.aan3377](https://doi.org/10.1126/science.aan3377)).

To address each of these papers separately:

- [10.1002/anie.201402973](https://doi.org/10.1002/anie.201402973):

This is a very interesting finding and can be investigated using our model with minimal modification. Thank you for mentioning this study.

- [10.1038/s41467-019-09002-6](https://doi.org/10.1038/s41467-019-09002-6):

This concept of minimal local frustration is very compelling and may indeed help to explain the processes that we observed. It may be worth designing an analogous structure and simulating it using our software to investigate this effect.

- [10.1126/science.aan3377](https://doi.org/10.1126/science.aan3377):

This observed effect is likely quite sensitive to excluded volume; investigating it might require more careful parameterization of the excluded volume behavior especially around junctions. In addition, it is likely that staple base stacking is an important driving force to this transition, so adding these interactions to our model would be necessary to investigate this information relay effect.

We have discussed these points briefly in the discussion section (page 23).

sequence-dependence, hysteresis and isomerization are important phenomena in protein folding and reconfiguration. Hence, I imagine that using this model to simulate the folding of some representative protein candidates could be very instructive and also help revealing other aspects of DNA origami folding in general.

Thank you for your comment. We believe that this process may indeed bear many similarities to protein folding and would like to implement a similar model in proteins as well to reveal more about both protein folding and DNA origami folding.

a last point: can the authors clearly state the size and MW of the structures they have simulated? which is the range of sizes and MW that can be reasonably simulated by their approach?

The sizes of the structures that we simulated, in order, are:

- sheet structure: 768 nucleotides / 500 kDa.
- all 4HB structures: 3,584 nucleotides / 2.33 MDa.
- 32HB structure: 8,192 nucleotides / 5.32 MDa.

These values have been placed in the caption for each figure where the design is introduced. With enough allocated resources, it should be feasible to simulate the folding of full-sized DNA origami (on the order of 15,000-16,000 nucleotides, 10 MDa) in less than a month on a modern, fast CPU with ≥ 12 cores. Additional optimization may enable even larger structures to be simulated. We will note that we are not experts in high performance simulation coding, so our simulation program is most likely operating well below its optimal performance.

Response to Reviewer 3:

DeLuca *et al.* describe a coarse-grain model that provides access to information about the assembly of DNA origami structures. Interaction between staple and template is modeled at a domain level where each domain represents 8 nucleotides. The model uses a switchable force field to capture the mechanical properties of single-stranded and double-stranded domains. The model is first benchmarked against oxDNA which has been used extensively to characterise folded DNA origami structures. It is then used to explore isothermal assembly pathways of (small) origami systems.

The model is a welcome addition to the tools available for modelling DNA origami assembly and potentially allows end users to begin to design not just final assembled structure but assembly pathway too. That said, its adoption will depend upon ease of use (note the value of oxDNA.org and oxView in opening access to new users by lowering the barrier to adoption).

We thank the reviewer for their positive feedback and for helping us to improve our manuscript. We agree that ease of use is a critical factor in the eventual adoption of this software. We have accordingly made significant modifications to the source code to make it more user friendly, and included a user guide with example folders in the GitHub repository (the user guide is hosted separately at <https://daniel-duke.github.io/dnafold-docs/>). These usability improvements include:

- Modifying the parameter ingest to take an input file in .txt format with parameters specified in key=value format so that the software only needs to be compiled once (we have included documentation and examples to make it easy to formulate these input files).
- Reducing the number of arguments required to run a simulation to two: the executable and the input, e.g. “./dnafold input.txt”.
- We have prepared a detailed user manual to increase the usability of the code. The manual contains three sections: (1) installation and running instructions; (2) descriptions of all the parameters, their units, and their defaults; and (3) a description of how to create compatible caDNA designs. The last section contains several examples of designs that DNafold would successfully simulate, as well as designs that would throw flags or produce undefined behavior.
- We have improved the JSON ingest behavior to automatically format the topology file so that the topology and trajectory can be opened using the Open Visualization Tools (OVITO) package or any package that reads LAMMPS formatted trajectories, thereby allowing the visualization all of the individual DNA strands and their bonds.
- Simulations now generate a simulation metadata file that documents the value of every user-set parameter in the simulation.

We now feel confident that a user will be able to independently run our code to simulate their own DNA origami designs with parameters of their own specification. We note that we are continuing to improve this software and intend to expand the available features and sophistication of the model. Additionally, we request that users compile and run this software on MacOS, Windows Subsystem for Linux, or Linux operating systems. Building the software in a native Windows environment is not currently supported. In the long-term, we would like to integrate this model and software into the existing ecosystem of webapp-based tools in the DNA nanotechnology field.

Do simulations contain more than one copy of each staple? Staple excess is an experimentally important though poorly understood parameter and one that models tend not to capture well (staple excess seems to facilitate folding in experiment and trap misfolds in simulation - likely due to effectively irreversible interactions at simulation temperature).

We thank the reviewer for raising this important point. Our model currently employs only one copy of each staple – the effect of staple-to-scaffold ratio on folding behavior will be investigated in the future to

gain a better understanding of this important fabrication parameter. **This point has been emphasized in the discussion section (see page 24).** Our very preliminary simulations indicate that the collapse mechanism of 4HB structures is preserved when a 10:1 staple-to-scaffold ratio is used. Since the free energy change of hybridization ΔG_{hyb} is large, the escape time for these misbinding events is correspondingly very long, not only because of large boxes with a large amount of diffusion but also because of the escape times of these defects which the reviewer has described (these escape times are short on the scale of experiments but long on the scale of simulations).

Are staple binding events readily reversible at the simulation temperature?

Yes, staple binding events are reversible. The reversal of binding occurs at a rate dictated by the hybridization strength (melting barrier) of 10 kcal/mol per bead. This typically means that unbinding does not occur often unless there is some stress on the structure in which case the staples will be readily pulled off of the structure. Since the hybridization potential was derived using umbrella sampling of the accurate oxDNA coarse-grained model (see Methods: hybridization potential), we believe that the potential is capturing the correct behavior. We are not sure if our model is capable of accurately capturing the intricacies of effects like toehold-mediated strand displacement, but this is something we are actively considering. **We have added a small clarification about this point on page 8 of the manuscript.**

What effect does the temperature have on assembly, how was the simulation temperature decided, how close is it to the melting temperature of the structures under study?

The hybridization potential was parameterized at 300 K. More work is required to describe the temperature dependence of this potential because of vibrational entropy terms that we do not capture with our parameterization and DNA resolution. Calculating the true melting temperature would require parameterizing based on a model that better captures the vibrational entropy at all relevant temperatures, and then constructing a 2D hybridization potential between coarsened beads $U_{\text{hyb}}(d, T)$ where T is the temperature of the system and d is the separation between the coarsened beads. This would enable accurately modeling behavior like duplex melting and origami unfolding with increasing temperature.

I'm not sure that I understand what is going on during the two-stage assembly. It is odd that the average staple behaviour is discussed when information about individual staples is available (e.g. as presented in Fig 5C). It would be worth investing time into how to present these results. I would like to be able to better understand the order in which contacts are made (for example contact maps colored by domain incorporation time).

We thank the reviewer for their comment. The 4HB folding process is highly random and so the order in which contacts are made is correspondingly mostly random. This is discussed in Schneider *et al.*, Sci. Adv. 2018 (doi.org/10.1126/sciadv.aaw1412), where single-staple resolution experiments revealed that the folding pathway was relatively indeterminate. **For reference, we provide further below plots of binding times for ten individual simulations (Figure 1).** It is difficult to make additional conclusions from these plots; perhaps something more could be gathered with more of such charts where significant statistics beyond those discussed in the manuscript (*i.e.*, elevated average binding time of the large staples) might arise. We also originally had contact maps but found them more difficult to interpret.

Something has gone wrong with the plots in 5C. It would help to have a little white space between domains, to make sure that domains line up in columns (perhaps that is what was intended - it looks a little like two such plots might have been overlaid)

We thank the reviewer for their comment. **We have re-drafted this figure panel and hope that it is clearer.**

Figure 1: Computed binding times for ten individual Brownian dynamics simulations of the 4HB structure with ΔU_{bind} of 10 kcal/mol.

REVIEWERS' COMMENTS

Reviewer #1 (Remarks to the Author):

I thank the authors for addressing all my questions.

I am in favor of publishing the article.

Reviewer #1 (Remarks on code availability):

The provided examples are running well.

And the suggested visualization tool is working.

The tool is well documented.

Reviewer #2 (Remarks to the Author):

The authors have addressed all my points carefully and completely. I support publication of this work in Nature Communications.

Reviewer #3 (Remarks to the Author):

I am happy that the authors have addressed the comments made in all reviews and am glad to see that they have improved the accessibility of the code. I recommend publication and wish the authors success in extending their model to look at strand displacement and misbonded interactions.

Reviewer #3 (Remarks on code availability):

I was able to install and run the code following instructions in readme.md

Response to Reviewer #1:

I thank the authors for addressing all my questions. I am in favor of publishing the article.
Remarks on code availability: The provided examples are running well. And the suggested visualization tool is working. The tool is well documented.

We sincerely thank the reviewer for taking the time to review our manuscript and for providing useful comments and suggestions for further improvement of the manuscript. We are happy to hear that the reviewer appreciated our revised manuscript and recommended it for publication.

Response to Reviewer #2:

The authors have addressed all my points carefully and completely. I support publication of this work in Nature Communications.

We sincerely thank the reviewer for taking the time to review our manuscript and for providing useful comments and suggestions for further improvement of the manuscript. We are happy to hear that the reviewer appreciated our revised manuscript and recommended it for publication.

Response to Reviewer #3:

I am happy that the authors have addressed the comments made in all reviews and am glad to see that they have improved the accessibility of the code. I recommend publication and wish the authors success in extending their model to look at strand displacement and misbonded interactions.

Remarks on code availability: I was able to install and run the code following instructions in readme.md

We sincerely thank the reviewer for taking the time to review our manuscript and for providing useful comments and suggestions for further improvement of the manuscript. We are happy to hear that the reviewer appreciated our revised manuscript and recommended it for publication.